# Nucleation phenomena and extreme vulnerability of spatial k-core systems

Leyang Xue [1,2,3], Shengling Gao[3,4], Lazaros K. Gallos [5] ✉, Orr Levy [6,7], Bnaya Gross [3], Zengru Di [1,2] ✉ & Shlomo Havlin [3] ✉

K-core percolation is a fundamental dynamical process in complex networks with applications that span numerous real-world systems. Earlier studies focus primarily on random networks without spatial constraints and reveal intriguing mixed-order transitions. However, real-world systems, ranging from transportation and communication networks to complex brain networks, are not random but are spatially embedded. Here, we study k-core percolation on two-dimensional spatially embedded networks and show that, in contrast to regular percolation, the length of connections can control the transition type, leading to four different types of phase transitions associated with interesting phenomena and a rich phase diagram. A key finding is the existence of a metastable phase where microscopic localized damage, independent of system size, can cause a macroscopic phase transition, a result which cannot be achieved in traditional percolation. In this case, local failures spontaneously propagate the damage radially until the system collapses, a phenomenon analogous to the nucleation process.

A phase transition is characterized by changes in the macroscopic properties of a system when crossing the critical point[1,2]. Percolation is a mathematical model that has been widely explored to exhibit and understand geometric phase transitions and explain the conditions under which these transitions are discontinuous (first-order) or continuous (second-order)[3,4]. Of particular recent interest are systems that exhibit mixed-order transitions showing features from both first- and second-order transitions. Recent advances indicate that this behavior may be related to long-range interactions[5–8]. In recent years, mixed-order transitions have been reported for k-core pruning in random networks[8–12]. The k-core approach, which involves the iterative removal of nodes with a degree smaller than $k$ from a network, provides a unique perspective on the underlying structure of the network and its robustness. Notably, k-core pruning has been seen as a percolation process in which nodes are removed from the outer layers of the network, leading to the term "k-core percolation"[9,10,13,14]. K-core percolation has found widespread applications in real systems with spatial characteristics. Examples include the brain[15–17], cellular structures[18,19], the Internet[20], communication infrastructures[21,22], and ecological networks[23–25]. Furthermore, as in any percolation process, k-core percolation has expanded its reach to dynamic processes[9,10], emphasizing its versatility and relevance. At the heart of k-core percolation research lies the investigation of phase transitions and critical behaviors[11,12,26,27]. Intriguingly, k-core percolation on random networks has been shown to result in a mixed-order phase transition for $k \geq 3$[11,28], where the order parameter (the size of the giant component of the k-core) exhibits an abrupt jump akin to first-order phase transitions but near criticality it features a scaling behavior typical to second-order phase transitions.

While k-core percolation has received extensive attention and demonstrated mixed-order transitions, it mainly focuses on non-spatial random networks. In reality, numerous real-world systems, from transportation networks and power grids to communication systems and brain networks, are commonly embedded in two- or three-dimensional space[16,29–31]. Despite their widespread existence, our

[1]International Academic Center of Complex Systems, Beijing Normal University, Zhuhai 519087, China. [2]School of Systems Science, Beijing Normal University, Beijing 100875, China. [3]Department of Physics, Bar-Ilan University, Ramat-Gan 52900, Israel. [4]School of Mathematical Sciences, Beihang University, 100191 Beijing, China. [5]DIMACS, Rutgers University, Piscataway, NJ 08854, USA. [6]Department of Immunobiology, Yale University School of Medicine, New Haven, CT, USA. [7]Howard Hughes Medical Institute, Chevy Chase, MD, USA. ✉e-mail: lgallos@gmail.com; zdi@bnu.edu.cn; havlins@gmail.com

understanding of the robustness and vulnerability of these spatially embedded networks undergoing the k-core process remains very limited. As a result, a comprehensive framework for understanding the effects of spatial embedding is notably absent. Although the role of long-range interactions in mixed-order transitions was found to be pivotal[6,8], the exact mechanism through which the k-core induces this transition remains an open problem. In particular, when dealing with random networks featuring small-world characteristics, the lack of spatial, finite length scale connections within the k-core structure poses a significant challenge in examining how the characteristic length of the links influences the vulnerability of the system. By investigating k-core percolation in spatially embedded random networks, we can gain insight into how spatial constraints impact network functionality[30]. Currently, a limited number of studies have explored the spatial effect of k-core percolation, with a predominant focus on phase transitions and critical behavior[32,33]. However, these studies did not consider the effect of controlling the length scale of links which, as we show here, is critical for understanding the network vulnerability. Consequently, there is a discernible gap in the development of a unified framework to understand the underlying mechanisms that drive catastrophic breakdowns near critical points.

Here, we develop a comprehensive numerical framework for investigating the attributes of k-core percolation within spatially embedded two dimensional networks. We provide evidence, based on extensive simulations, that the distribution of link lengths in a spatial system undergoing k-core pruning will determine the nature of the phase transition and we elucidate the mechanisms driving the transitions in k-core systems showing four different types of phase transitions. Importantly, we show that there exists a new regime in the phase diagram, an extreme risky phase, i.e., a metastable phase, where a microscopic intervention above a certain size anywhere in the system yields a macroscopic phase transition represented by the collapse of the network. This microscopic intervention corresponds to removing just a few nearby nodes, where the damage size is independent of the system size. Due to a nucleation spread process, the result of this minimum removal is the destruction of the entire system, i.e. a collapse at the macroscopic level. This finding demonstrates a fundamental vulnerability in these systems, a result which cannot be observed in traditional percolation studies. Therefore, spatially embedded networks may collapse significantly more easily than previously thought, and special attention must be paid to increase their robustness and avoid catastrophic damage.

More precisely, our primary focus lies on examining the impact of link length within a 2D spatially embedded network. Our objective is to discern the influence of the characteristic link length (denoted as $\zeta$) on the properties of the critical phase transition of the k-core. Small values of $\zeta$ favor links to nearby nodes, whereas larger values of $\zeta$ introduce increasingly longer-range links, and eventually, when $\zeta$ is of the order of the linear size of the system, $L$, one obtains a fully random network. We find that the percolation critical point $p_c$ for k-core percolation reaches a maximum value at a critical characteristic length $\zeta_c$, which depends on $k$. Below this value of $\zeta_c$ the phase transition is continuous, but becomes abrupt for $\zeta > \zeta_c$. Interestingly, the mechanisms behind the abrupt first-order transitions observed for $\zeta \geq \zeta_c$ are found to depend on the value of $\zeta$. For values of $\zeta$ that are slightly higher than $\zeta_c$, the transition is abrupt and results from the propagation of a spontaneous local failure, causing the radius of the damage to increase until the system fully collapses. This is similar to a nucleation process in the gas-liquid model or in the spin model. However, when $\zeta$ is much larger than $\zeta_c$ and of the order of system size, a critical branching process emerges as the nodes fail homogeneously anywhere in the system, at a constant low (microscopic) rate, eventually leading to an infinite-size avalanche. The analysis of the critical conditions and the underlying mechanisms reveals a rich phase diagram. Of particular interest is that in addition to the above phases, we observe an extremely vulnerable phase, i.e., a metastable phase where a microscopic localized attack anywhere in the system above a critical radius $r_h^c$ (which is independent of the system size) spontaneously spreads and leads to the full collapse of the system. These observations point to inherent extreme vulnerabilities of the system and can provide a comprehensive understanding of the mechanisms that lead to continuous second-order, mixed-order, and first-order transitions.

The extreme vulnerabilities of spatially embedded k-core network systems highlight the necessity to take into account the characteristic length of links when designing robust spatial networks. Furthermore, our insight about the microscopic processes and their origin during the mixed-order and first-order abrupt transitions in k-core networks could shed light on the mechanisms of many systems where such transitions occur.

## Results
### The model
The structure of a spatially embedded network is largely determined by the dimension of the embedding space, the location of the nodes, and the length of the links in this space. For example, a two-dimensional network where each node only connects to short-distance neighbors can be mapped to a square lattice, and all critical exponents of the percolation transition in the embedded network will be the same as in a pure square lattice, since they share the same spatial dimension and thus belong to the same universality class[34]. Here, to investigate the relations between the link length and the k-core percolation properties, we place all network nodes at the sites of a 2D square lattice. The number and length of links for each node are determined by (a) the chosen degree distribution and (b) the length distribution of the links. Given an average degree $\langle k \rangle$, we construct spatially embedded networks using the $\zeta$-model[29,31,35], which leads to a Poisson degree distribution. In the $\zeta$-model, the Euclidean length $l$ of the links between pairs of nodes follows an exponential distribution,

$$P(l) \sim e^{-l/\zeta}. \tag{1}$$

The parameter $\zeta$ defines the characteristic length of the links in the network. This model allows us to control the embedding strength of the spatial network by adjusting the value of $\zeta$. For $\zeta \to \infty$, the link length probability is independent of the distance, and the model becomes equivalent to a non-spatial random network. Thus, this model can also describe a random network in the limiting case when the $\zeta$ values are of the order of the lattice size. On the other hand, for values of $\zeta$ much smaller than the lattice linear length $L$, the probability of long-range links tends to zero, and this limiting case is now a 2D lattice structure. An example of a spatially embedded network generated by the $\zeta$-model is shown in Fig. 1a. The specific implementation of this model is described in the Methods section.

### k-core percolation
In the k-core percolation process, the initial step involves a random removal of nodes with a probability of $1 - p$. This initial removal is the only external intervention in the system, which then evolves spontaneously via cascading until reaching a steady state. Subsequently, we initiate the k-core pruning procedure, in which all nodes possessing degrees lower than $k$ are eliminated, and the degrees of the remaining nodes are updated accordingly. This step is repeated with the recalculated degrees until all remaining nodes have degrees equal to or greater than $k$. Figure 1a demonstrates a simple 3-core percolation process in a small spatially embedded network. In the results, we show the analysis using the 5-core percolation. For more percolation results at different $k$ values, see the Supplementary Information.

The fraction of nodes, $P_\infty$, in the giant connected component (GCC) plays the role of the order parameter in the phase transition. The GCC corresponds to the remaining k-core structure in the network,

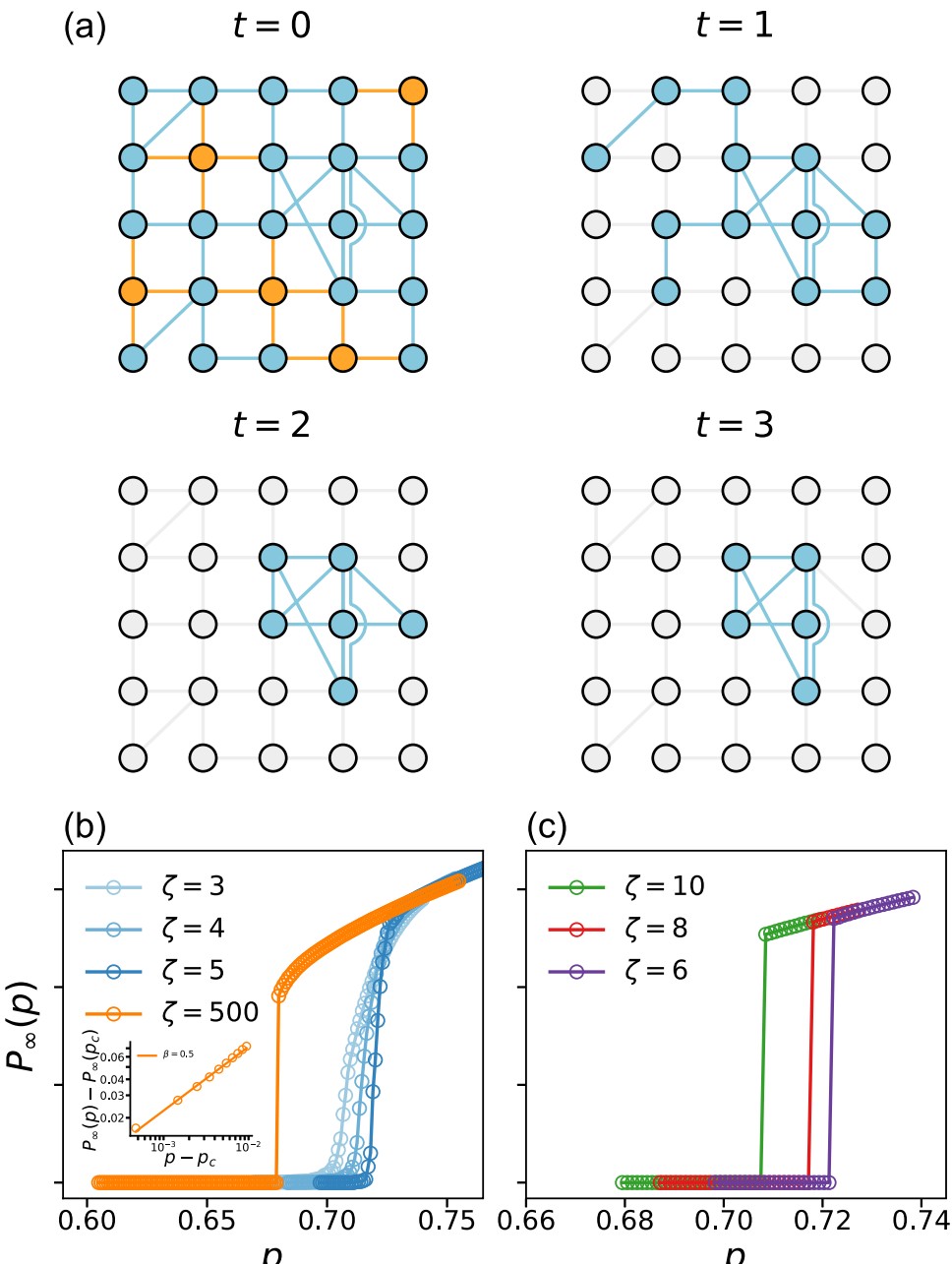

**Fig. 1 | K-core percolation on spatially embedded networks. a** Illustration of a 3-core percolation process on a 2D spatial network. The network is generated by the $\zeta$-model with $N = 25$, $\langle k \rangle = 3.6$ and $\zeta = 1$ (see Eq. (1)). At $t = 0$, we randomly remove 5 nodes (marked in orange), and all links connected to them also fail. In each subsequent step, $t$ of the pruning process, we remove all nodes whose degree becomes smaller than 3, until all remaining nodes have a degree equal to or larger than 3 at $t = 3$. **b, c** Size of the giant component $P_\infty$ in a 5-core percolation process for different $\zeta$ values as a function of the occupation probability $p$ (the fraction of unremoved nodes). The network size is $N = L \times L = 1000 \times 1000$ and $\langle k \rangle = 10$. The

results for continuous phase transitions are averaged over 10 realizations, while a single typical realization is used for the discontinuous phase transitions. For $\zeta = 500$, the network is practically completely random, while for $\zeta = 3$ it is similar to a 2D lattice network. Inset: Plot of the change in the size of the giant component as we approach the critical point $p_c$ in the 5-core percolation. The slope of the line, 1/2, corresponds to the $\beta$ exponent: $P_\infty(p) - P_\infty(p_c) \propto (p - p_c)^\beta$. Such a transition represents the critical regime and is called a mixed-order transition. Similar results for other k values are shown in the Supplementary Information.

and characterizes the robustness of the system, which we assume to be functional only if the remaining fraction of the GCC has a measure larger than zero. The critical probability $p_c$ is defined from the plot of $P_\infty$ vs $p$ as the value of $p$ where $P_\infty$ vanishes for the first time as we lower $p$.

Damage propagation through a k-core percolating process can be demonstrated in a simple model of a two-dimensional square lattice. In terms of the $\zeta$-model described above, this lattice corresponds to the limiting case of $\zeta = 1$ and $k = 4$, with the absence of randomness in both

$\zeta$ and $k$. All nodes in this lattice have initially a degree $k = 4$. If we consider a 4-core percolation process, then the removal of a single node anywhere in the lattice will lead to the subsequent removal of its four neighbors, that will now have degree $k = 3$. The neighbors of these nodes will also be removed, since their degrees become also 3 or 2, creating a damage propagation front with nodes of degree less than 4, which will keep expanding and will eventually destroy the entire lattice system. We discuss further this simple model in the Supplementary Information.

## The effect of the characteristic link length

In this section, we show that the system undergoes a phase transition as we increase the initial fraction of removed nodes, $1 - p$, for a fixed value of the characteristic link length, $\zeta$. The phase transition is determined by the vanishing value of $P_\infty$ at equilibrium, i.e., at the end of the k-core percolation process, see Fig. 1. The nature of this transition changes as we increase the $\zeta$ value: (a) for small $\zeta$ of order of a few unit lengths, we observe a continuous transition, (b) for intermediate values of $\zeta$ the transition becomes discontinuous, while (c) for large values of $\zeta$ the transition becomes mixed-order. As we show below, each type of transition is associated with a different mechanism of damage propagation, which corresponds to (a) fractal disintegration, (b) nucleation and propagation of a single hole, and (c) random locations of cascading failures. We detect these mechanisms by following the evolution of the k-core percolation cascading process after the initial node removal.

We study the behavior of the phase transition for the 5-core percolation, as we vary the characteristic length $\zeta$. For $\zeta = 3$, the network exhibits strong spatial effects, with connections largely restricted between neighboring nodes. This system undergoes a continuous phase transition at $p_c$, as shown in Fig. 1b. This is in contrast to the discontinuous transition for the case $\zeta = 500$ in the same plot, which behaves, as expected, similarly to the percolation of k-core in random networks[9]. Due to the absence of long-range links for small $\zeta$ values, the damage remains localized and is confined to the vicinity of the randomly removed nodes, leading to the absence of global or catastrophic failures. As we increase the value of $\zeta$, we introduce longer links that result in both a change in the critical value $p_c$ and a change in the nature of the transition that becomes abrupt (Fig. 1c). It is important to note the differences between the abrupt behavior for $\zeta > 6$ but close to 6 (Fig. 1c) and the abrupt transition for $\zeta = 500$ (Fig. 1b). While for $\zeta = 500$ one can see in a $P_\infty(p)$ a curvature just above $p_c$, representing a singular behavior with a critical exponent $\beta = 1/2$ (see the inset of Fig. 1b), for $\zeta$ just above 6 there is no such a singular regime and the drop in $P_\infty$ when decreasing $p$ is abrupt. This feature already indicates, as we show later, that these two transitions have different mechanisms.

In Fig. 2a we plot the value of $p_c$ as a function of $\zeta$. The value of the critical point increases with $\zeta$ until it reaches a maximum value at $\zeta_c = 6$. For $\zeta$ below $\zeta_c = 6$, larger $\zeta$ values enable the propagation of failures over a large area of the network. As a result, a smaller number of initial removed nodes is needed to cause system collapse, yielding for larger $\zeta$ a larger value of $p_c$. In this range, below $\zeta_c$ the phase transition remains continuous (Fig. 1b), indicating that failures do not propagate far enough to trigger an abrupt collapse. As a result, at $p_c$ some areas of the network sustain limited damage while others are impacted more significantly. For example, the top left panel in Fig. 2a shows the resulting GCC at the end of the k-core percolation process for $\zeta = 4$, which exhibits clear fractal features at $p_c$, similar to percolation in a single lattice network[34]. This is in marked contrast to the resulting structure of the GCC of a random network at $\zeta = 100$ where, just before collapsing, the damage is spread uniformly throughout the network (top right panel in Fig. 2a).

For $\zeta$ values above $\zeta_c = 6$ and smaller than the system length, the phase transition becomes discontinuous, as seen in Fig. 1c, but the transition behavior is different for values close to $\zeta_c$ compared to much larger values of $\zeta$ of the order of system size, $L$. When $\zeta$ is above $\zeta_c$ but of the order of $\zeta_c$, the k-pruning now results in a drastically different mechanism, as shown in the bottom row of Fig. 2a. After the random removal of the initial nodes, a large enough hole emerges spontaneously somewhere in the network, due to the random spatial disorder of nodes and links in the system. Once this hole emerges somewhere in the system the $\zeta$ links enable its propagation. This yields that the front of the hole spontaneously expands outwards until it consumes the entire network. Note also the simplified demonstration in the Supplementary Information.

This process is analogous to the well-known nucleation process observed during the freezing of water. This is one of the key aspects of this transition which we discuss extensively later in this paper. Hence, there is no scaling behavior for $P_\infty$ near criticality, similar to a first-order phase transition, see Fig. 1c. The visualization of the cascading process for $\zeta = 10$ at the bottom row of Fig. 2a clearly demonstrates this interpretation. Initially, random fluctuations in network density create a region where the fraction of remaining nodes is significantly lower than the overall average, thus creating a hole in the system. The surviving nodes remain largely connected and the value of $P_\infty$ remains high, since only small clusters are isolated from the GCC. However, as the hole evolves and propagates, it eventually leads to the sudden collapse of the entire system. Further supporting explanations can be found in Supplementary Figs. S6 and S8.

As $\zeta$ increases above $\zeta_c$, the value of $p_c$ gradually decreases and approaches the asymptotic value observed in random networks, $p_c \approx 0.6798$, as seen in Fig. 2a. When $\zeta$ becomes very large, of the order of the system length, the transition becomes mixed-order (Fig. 1b), in contrast to pure first-order transitions at intermediate values of $\zeta$ (Fig. 1c). In this mixed-order transition, for large $\zeta$, the order parameter $P_\infty$ still exhibits an abrupt jump, but $P_\infty$ shows a scaling behavior close to $p_c$ with a critical exponent $\beta = 0.5$, similar to what has been found in a random network[9], see inset in Fig. 1b. In this case, the characteristic lengths are of the order of the system linear size and a hole cannot propagate radially. Hence, there is no nucleation process and the system is driven by a bifurcation process, which has been found in ref. 10. As shown at the top right panel of Fig. 2a, the network is highly homogeneous near criticality, without allowing the formation of any hole of significant size.

The speed of the damage spread depends on the characteristic link length, as shown in the slopes of Fig. 2b. Once the hole size reaches a critical finite size, the hole radius exhibits a linear growth, associated with the nucleation process, which continues until the hole reaches the system boundary. Longer link lengths, $\zeta$, yield faster propagation of the damage since the k-core process will affect neighborhoods at longer distances. As the link length increases, fewer steps are needed to form holes with a critical finite size, and the duration of the nucleation process is shorter, so that the hole grows faster than in smaller $\zeta$.

The above results are based on 5-core percolation, but similar behavior is found for other $k \geq 3$ core percolation, see Supplementary Figs. S4 and S5. Higher values of $k$ exhibit stronger cascading effects that lead to a decrease in $\zeta_c$, as shown in the inset of Fig. 2a. For $k < 3$, the phase transition remains continuous at $p_c$ independently of the value $\zeta$, because there is limited propagation of failure in 2-core and 1-core percolation[8], which is an effect of percolation rather than spatial embedding. However, for $k \geq 3$, the characteristic length of the links leads to the rich dynamical phenomena as described above.

In summary, the type of phase transition at different characteristic lengths is determined by the nature of spontaneous fluctuations following the node removals in the system. For links of short lengths, these fluctuations only act locally, creating progressively all sizes of gaps in a typical fractal fashion. When the characteristic lengths of $\zeta$ are very long, the system fails uniformly, since any node removal can impact nodes at any distance. In the intermediate range, a spontaneous emergence of a localized hole which eventually consumes the entire system leads to a rapid system disintegration, with clear features of a first-order transition.

## The dynamic process of critical cascading failure

In the previous section, we described how the value of $\zeta$ controls the type of transition and indicated that there exist two distinct mechanisms that can lead to discontinuous transitions when $\zeta > \zeta_c$. In this section, we focus on studying and quantitatively demonstrating these mechanisms that yield the cascading failure process for $\zeta = 7$ and for $\zeta = 500$, near and far from criticality (Fig. 3). Here, we remove a fraction

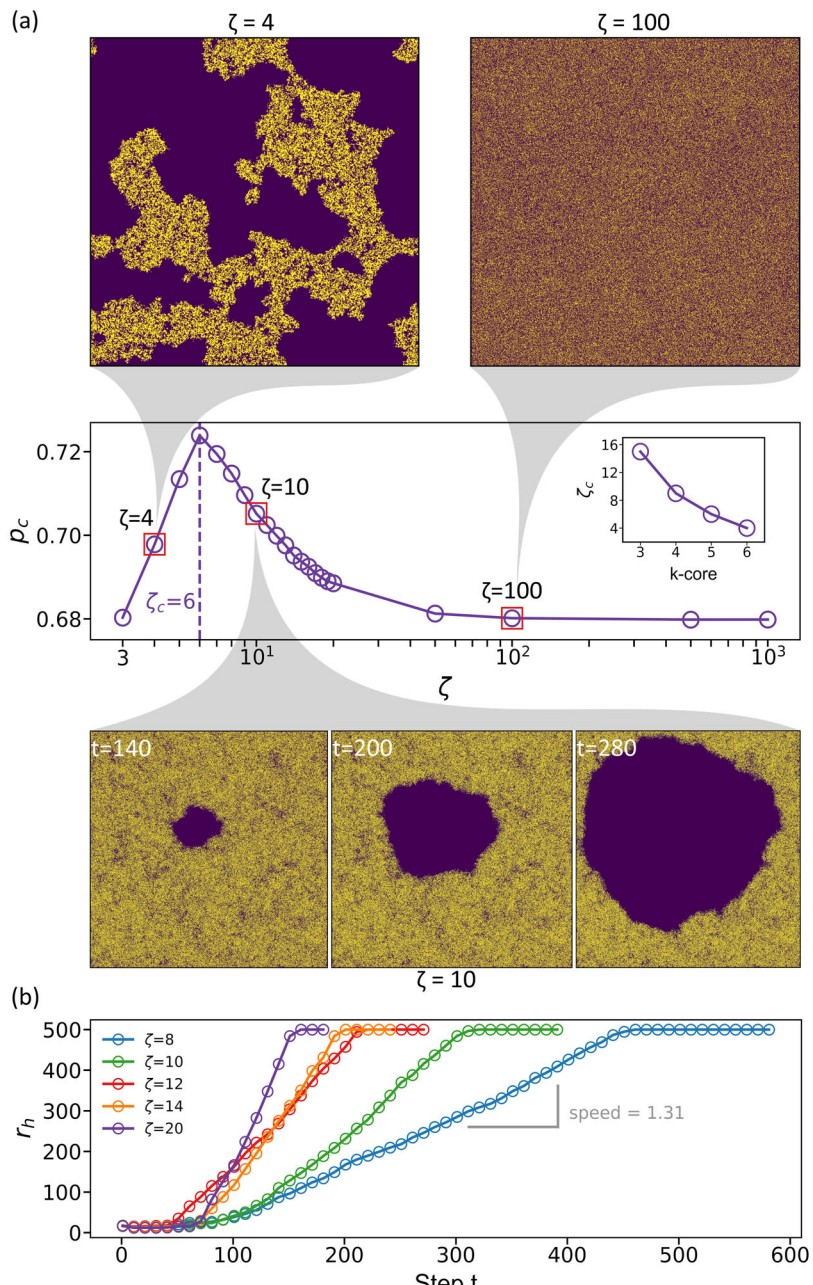

**Fig. 2 | Variation of the link length leads to three different transitions for k-core percolation on spatial networks. a** The critical point $p_c$ as a function of the characteristic link length $\zeta$ for 5-core percolation. The results are averaged over 50 realizations. The maximum value of $p_c \approx 0.724$ at $\zeta_c = 6$ also indicates the point where the transition shifts from continuous ($\zeta < \zeta_c$) to discontinuous transitions ($\zeta > \zeta_c$). Note that similar qualitative behavior is expected to occur for any k-core where $k \geq 3$. Inset: $\zeta_c$ plotted as a function of the value of $k$ used for k-pruning. The two panels in the top row show the stable giant component of two spatial networks with $\zeta = 4$ and $\zeta = 100$ at $p_c$, just before collapsing, which corresponds to the points marked by red squares in the plot. The panels at the bottom row illustrate the evolution of the giant component of a spatial network with $\zeta = 10$ at $p_c$. This figure shows the spontaneous way of the abrupt collapse in Fig. 1c. The three snapshots show the giant component at times $t = 140$, 200 and 280 steps of iterations where a growing hole is evident. In the above, gold and purple dots indicate surviving and removed nodes, respectively. The network size here is $N = 1,000,000$ and the average degree $\langle k \rangle = 10$, is the same as in Fig. 1. **b** Evolution of the hole size, $r_h$, with time $t$ for a 5-core percolation. As $\zeta$ approaches $\zeta_c$, the nucleation process lasts longer, and the critical branching process becomes shorter.

$1 - p$ of nodes as a starting point, similar to Fig. 1b, c, and then follow the evolving giant connected component $P_\infty(t)$ in each iteration (time step), see Fig. 3a, b. Notice that 'time' is only used here as a convenient artificial metric to follow the system evolution, and in principle one could assume that nodes fail immediately when their current degree falls below the k-core threshold, so that the initial node removal spontaneously leads to the final state of the system. However, it is convenient to define each iteration of removing nodes as a single time-step, so that we can follow the microscopic changes in the system during the k-core pruning process and we can identify the microscopic mechanism which drives the system to this final steady state.

To clearly differentiate between the two dynamical processes for $\zeta \geq \zeta_c$, besides following the giant component we also track the largest cluster formed by failed nodes up to the time step $t$, $P^f_\infty(t)$, in Fig. 3c, d. We also monitor the branching factor, $\eta_t$ defined as the ratio between failure sizes at two successive time steps $t$ and $t - 1$ in Fig. 3e, f.

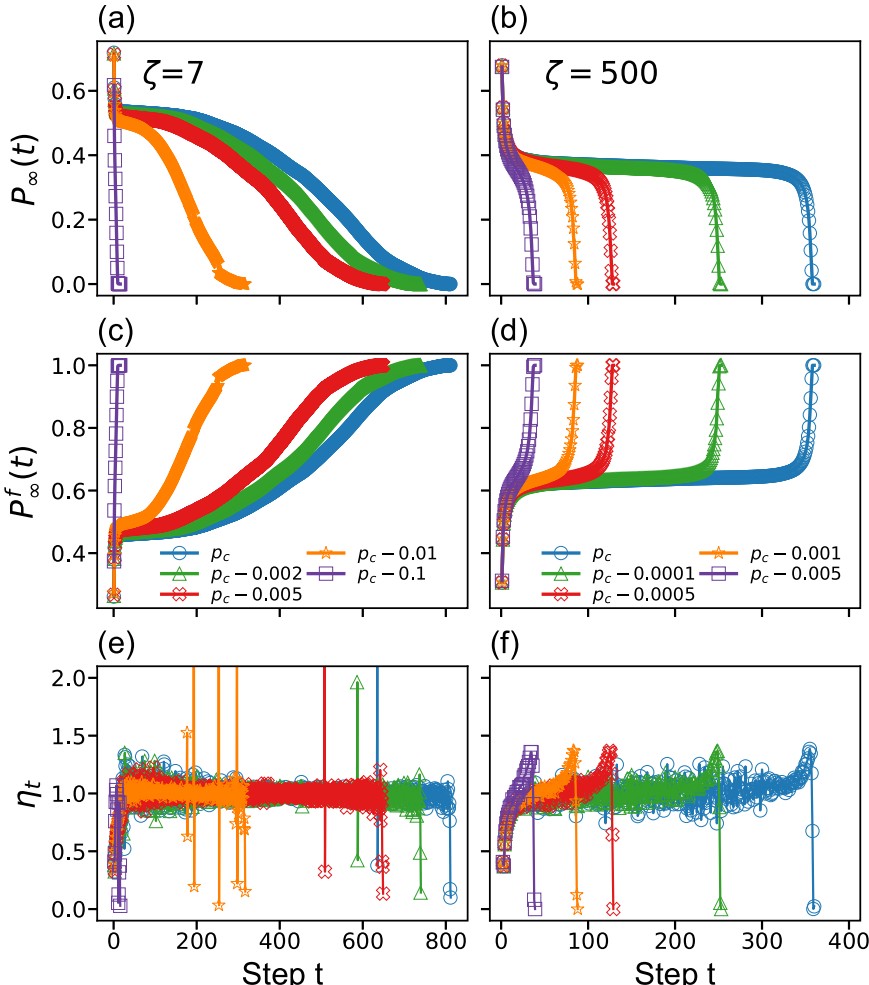

**Fig. 3 | Evolution of cascading failures in k-core percolation: two distinct processes and mechanisms.** The left column, **a**, **c**, and **e** corresponds to $\zeta = 7$, and the right column, **b**, **d**, **f**, corresponds to $\zeta = 500$. Different colors represent the values of $p$, as we get closer to criticality $p_c$ where the length of the plateau increases. **a**, **b** The size of the giant component $P_\infty(t)$ for a fixed $p$, as a function of the k-pruning time step $t$. $P_\infty$ for both cases exhibits a plateau followed by **a** a slow parabolic decrease while in **b** a very sharp collapse. The parabolic decrease is a result of the nucleation process seen at the bottom row of Fig. 2a, in which the radius of damage increases close to linearly with $t$. **c**, **d** The size of the largest cluster formed from the failed nodes up to the time step $t$, $P_\infty^f(t)$. **e**, **f** The branching factor $\eta_t$ defined as $\eta_t = \frac{s_t}{s_{t-1}}$ where $s_t$ denotes the number of failed nodes in step $t$. In both cases, we can see that the branching factor is mostly close to 1 which characterizes criticality during the spontaneous cascades. All networks have the same size $N$ and average degree $\langle k \rangle$ as in Fig. 1. The plots shown are consistent with and support the concept of two types of transition shown in Fig. 2a, for small $\zeta$ and for large $\zeta$ values.

The analysis of these properties for $\zeta = 7$ highlights three main stages of distinct behavior. $P_\infty$ undergoes a rapid decline, followed by a plateau, then decreases parabolically and slowly to zero, as seen in Fig. 3a, in contrast to the case of $\zeta = 500$ in Fig. 3b, where $P_\infty$ shows a plateau followed by a sharp drop. This contrast can be understood by following the size of the largest failure cluster in Fig. 3c, d. While for $\zeta = 7$ the increase in radius is linear and the area parabolic, characterizing nucleation growth, for $\zeta = 500$, the nearly flat curve is followed by a fast increase, consistent with homogeneous failures followed by a nearly abrupt collapse. These differences in the evolution of failure propagation distinguish the pure first-order transition at values $\zeta_c < \zeta \ll L$ from the mixed-order transition at large $\zeta$ of the order of $L$. Note that Fig. 3e, f show that during the plateau the branching factor $\eta$ is close to 1 showing another critical feature. This cascading behavior found at $p_c$ is similar to those observed for $p$ values just below $p_c$, that is, at the area under the curve in Fig. 2a. For these values, one can see a shorter duration of the plateau as we move further from the critical curve, since the system is more fragile, as seen in Fig. 3a–d.

In short, Fig. 3 provides evidence that there are two distinct behaviors in the range of $\zeta$ above $\zeta_c$, where cascade failures follow a different path to network destruction.

## Localized microscopic attack: an extremely vulnerable metastable phase

We have shown above that for intermediate values of $\zeta$, spontaneous fluctuations are the main cause leading to network collapse via nucleation at the critical point $p_c$. Note that the regime above the curve of $p_c$ in Fig. 2a describes a state of existence of the network, that is the resulting GCC has a non-zero measure. However, we find here that within this area above the $p_c(\zeta)$ curve, there is a new extremely vulnerable phase where a localized microscopic attack anywhere in the system will trigger nucleation growth and network collapse. Importantly, the critical localized size of the attack does not depend on the size of the network. We call this phase a metastable phase, and we highlight this phase in Fig. 4a within the dashed line. The system in this phase behaves in marked contrast to regular percolation, where the network in the regime above $p_c$ always remains connected and cannot collapse by any localized attacks[36].

K-core percolation studies focus mainly on random removal of nodes. However, in spatially embedded networks and for the intermediate range $\zeta_c < \zeta \ll L$, there are strong correlations within the neighborhood of a node, due to the length scale $\zeta$. This is the main driving force behind the metastable phase in which networks become

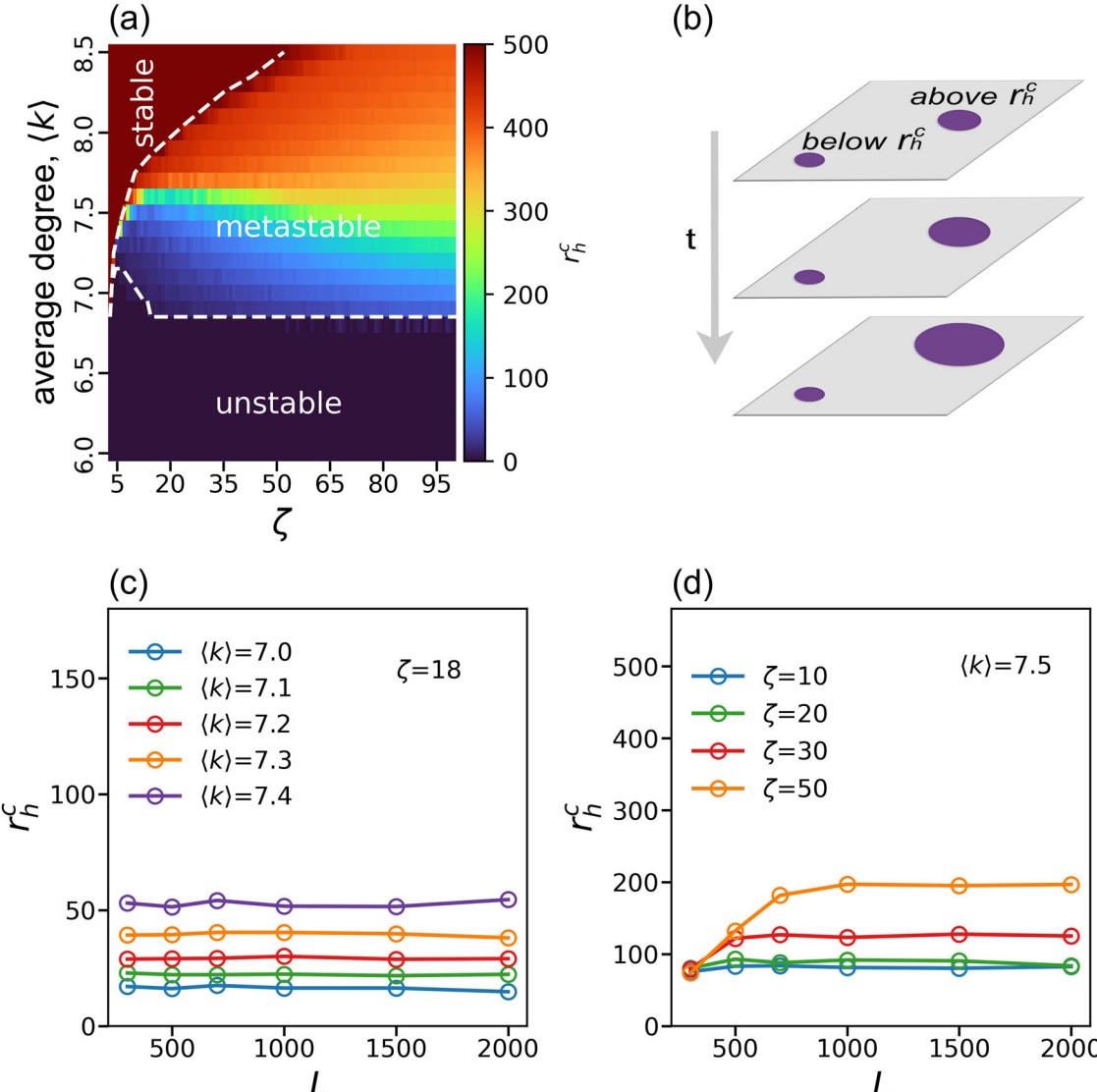

**Fig. 4 | A novel metastable phase that yields a phase transition via a microscopic damage. a** Phase diagram of spatial 5-core percolation as a function of $\zeta$ and $\langle k \rangle$. Purple is the unstable phase, and brown is the stable phase where the system is resilient to localized attacks. The colors within the dashed line represent the metastable phase where a circular damage above a radius $r_h^c$ will cause the system to collapse. The value of $r_h^c$ separates the unstable phase ($r_h^c = 0$, purple), the stable phase ($r_h^c \sim L/2$, brown), and the new metastable phase ($0 < r_h^c < L/2$, blue to orange). The size of the network is $N = L \times L = 1000 \times 1000$. **b** Illustration of the propagation of damage initiated from localized attacks with a hole of radius $r_h$ in the spatial

network. The hole on the left is smaller than the $r_h^c$, and the damage does not propagate, thus maintaining the same size as the initial state. However, damage caused by holes larger than $r_h^c$ on the right evolves and spreads radially throughout the system over time $t$. **c** Dependence of $r_h^c$ on the size of the system $L$ for $\zeta = 18$ and varying $\langle k \rangle$ values. The fact that $r_h^c$ does not increase with $L$ demonstrates that the localized damage is of microscopic size. **d** The dependence of $r_h^c$ on the system size for $\langle k \rangle = 7.5$ and varying $\zeta$ values. In **c** and **d**, each point represents the average of 10 realizations for $L < 1000$, and the average of 5 realizations for $L \geq 1000$.

extremely vulnerable to localized attacks, when nodes are removed from a small microscopic area in the network. Next, we study the effect of microscopic localized attacks on k-core percolation in spatially embedded networks via the artificial creation of a hole of radius $r_h$ in the network. In the metastable phase we show the existence of a critical hole radius $r_h^c$, below which the network is resilient and sustains damage, and above which the network collapses spontaneously.

We model a localized attack by creating a hole centered on a random node and removing all nodes located within a geometrical radius $r_h$ from this node and their links. This initial removal is then followed by the k-pruning process, which leads to further damage to the network. The cascade failures triggered by a localized attack depends on the k-core value, the characteristic length of the link $\zeta$ and the average degree $\langle k \rangle$ of the network. In our simulations, we systematically scanned the entire range of $\zeta$ and $\langle k \rangle$ for a given k-core and

found that the phase diagram of this system includes three distinct phases: stable, unstable, and metastable, see Fig. 4a. The stable phase is characterized by the containment of damage within the initial affected area, with no propagation of failure, independent of the size of the initial hole, i.e. only $r_h^c = \infty$ will trivially destroy the network. In the unstable phase, the failure propagates spontaneously throughout the system, even for a minimal attack ($r_h = 1$), that is, $r_h^c = 0$. In the metastable phase, the final state of the system depends on the initial hole size. If the hole radius is less than a critical finite size $r_h^c$, then the failure remains local, but for $r > r_h^c$ the localized damage propagates outwards spontaneously and the entire system collapses, as demonstrated in the schematic Fig. 4b.

A key feature of the metastable phase is that the value of the critical hole radius is independent of the system size $L^2 = N$, in contrast to random attacks where the collapse of system occurs when the

number of removed nodes is proportional to $N$. This means that a microscopic intervention yields a macroscopic phase transition. Indeed, in Fig. 4c, we can see that the radius of the critical hole remains unchanged when increasing the system size for any $\langle k \rangle$ value in the metastable state. Similarly, in Fig. 4d we fix $\langle k \rangle$ and vary $\zeta$ in the metastable regime. We find that $r_h^c$ increases linearly with $L$, but eventually reaches a threshold where it remains constant despite further increases of $L$, thus highlighting the impact of finite-size effects in the system. The important conclusion is that increasing the size of the system does not improve its resilience against localized attacks.

The process leading to the metastable phase relies on the formation of the failure front around a hole of size $r_h^c$. After removing the initial hole, a fraction of the remaining nodes on the failure front may have remaining degrees below $k$ at typical distance $\zeta$. This leads to the subsequent removal of these nodes in the failure front, causing the propagation of damage to continue further radially outwards. The local network density and the length of long-range links are therefore the key factors in determining the probability that the neighboring nodes of the front will retain a degree greater than $k$, or if they will be removed. As a result, the relation between $\langle k \rangle$ and $\zeta$ is enough to determine the eventual state of the system. We verify that the existence of a metastable state is a generic property in these systems with similar results for different k-core percolation processes and system sizes in Supplementary Fig. S9.

In this section, we demonstrated that a localized attack which removes all nodes within a radius $r_h$ can result to a total system collapse, as long as this radius exceeds a critical value $r_h^c$. More importantly, the value of this critical value is independent of the system size implying an extreme inherent vulnerability of the system to local attacks, even in large-scale systems.

## Discussion

In this work, we have developed a comprehensive framework to investigate the behavior of k-core percolation within spatially embedded 2D networks. We find that for random node removal, the critical point and the nature of the mechanism behind the phase transition in k-core percolation are determined by the characteristic link length, $\zeta$, of the spatial network. Furthermore, we unveil a rich phase diagram with second-, first-, and mixed-order phase transitions, each associated with a different mechanism depending on the value of $\zeta$. These mechanisms correspond to a fractal system disintegration, nucleation, and spontaneous microscopic random cascades. In the case of large $\zeta$ values, the system becomes more homogeneous because the majority of the links are long-range and there is no nucleation, yielding mixed-order transitions.

A key finding in our study is the existence of a previously unexplored regime in the phase diagram, which constitutes an extremely vulnerable phase, which we call metastable, where microscopic intervention results in a macroscopic phase transition. Notice that this metastable region is defined according to the behavior of the system under a microscopic localized attack, i.e. when we remove nodes within a circle of radius $r_h$ in contrast to random removal. The metastable state is a result of the system structure, as generated by the values of $\langle k \rangle$ and $\zeta$. The stability of the system is determined by the effect of this localized attack. If the system collapses by the random removal of $1 - p$ nodes, then the system is unstable. If the system remains connected after any size of localized attack, then the system is stable. In this study, we find that for a certain region of $\langle k \rangle$ and $\zeta$ the system becomes metastable, so that the size of the initial hole determines whether the system remains connected or if it collapses. We find that the critical size of the hole, $r_h^c$, above which the failure propagates and the system collapses, is independent of the system size and therefore can be regarded as a microscopic intervention. The value of $r_h^c$ is found to depend only on $\langle k \rangle$ and $\zeta$, showing that the underlying mechanism is determined by the system topology only. After

generating the initial hole of size above $r_h^c$, the links of size $\zeta$ enable the propagation of the hole radially via cascading failures causing the whole system collapse.

These findings highlight the significant vulnerability of spatial k-core systems and point to the important role of the underlying mechanisms that can shape the organization and robustness of real-world systems with spatial characteristics, such as the brain[15].

These results are unique in the fields of network science and phase transitions, since finite perturbations within a limited area in spatial systems are typically contained and cannot damage the entire system. The specific feature in k-core percolation that makes this behavior possible, is that a node needs to maintain a minimum degree at all times to survive, but at the same time its neighbors need to also obey the same condition, i.e., even if a node remains unharmed it fails if the degree of its neighbors becomes lower than the threshold. The long-range links facilitate this interaction between a node and its neighbors and by tuning the length of these shortcuts we can modify the behavior of the transition. This behavior is reminiscent of results in interdependent networks[37] where the existence of two types of interactions, i.e. connectivity and dependency links, leads to a phase diagram where both first-order and second-order transitions are present, and localized attacks of microscopic size can also bring down the entire system[5,29,38,39]. The main mechanism in interdependent systems is a cascade of removals alternating between the two layers, facilitated by the dependency links. In the case of spatial k-core percolation, however, there exists only one type of connectivity links so an analogy between the two systems is not straightforward. At a conceptual level, however, in k-core percolation the survival of a node also depends on the survival of its neighbors, which may suggest that a node's dependence on other nodes, either implicit or explicit, is a key feature that can lead to novel percolation properties. For example, previous research has shown that the two models have the same critical exponents near the critical point in random non-spatial networks[8,40,41]. These observations lead us to conjecture that nodes interdependence, broadly defined, could be the underlying common characteristic of a novel universality class, which has not been observed until now. Such a universality class would include common critical exponents across different systems, which can describe percolation a) in interdependent networks, b) in spatial k-core systems, c) in theory and experiments on interdependent superconducting networks[42], and d) possibly in other – currently unknown – systems. Similar to classical percolation where a universality class depends only on the system symmetries and dimensionality, we pose the question of whether there is a class of network systems whose percolation properties and critical exponents may depend on features such as interdependence or two types of interactions, which may be universal but not yet determined. The existence of such a universality class remains currently unknown and further work is needed to clarify its possible existence and its properties. On the other hand, our insight about the microscopic processes and their origin during the mixed-order and first-order abrupt transitions in both k-core and interdependent networks could shed light on the mechanisms of many systems where such transitions occur.

## Methods
### The $\zeta$-model
In this model, we consider a network with $N = L^2$ nodes and average degree $\langle k \rangle$, where each node is a site in a 2D Euclidean lattice with integer coordinates $(x, y)$. We fix the value of $\zeta$ and generate $N\langle k \rangle / 2$ random numbers, $l$, from the distribution $P(l)$ in Eq. (1), which correspond to link lengths. We implement periodic boundary conditions that restrict these link lengths to $l \leq L/2$. For each number, we attempt to generate a link starting at a random vertex that approximates this length, $l$, according to the following method. For each link of length $l$, we randomly select a node $(x_0, y_0)$ and uniformly choose a node $(x_0 + \Delta x, y_0 + \Delta y)$ from the candidate set of nodes at a distance $l$ from

node $(x_0, y_0)$. Notice that integer solutions may not exist for all values of $l$, as can be seen from the equation $l = \sqrt{\Delta x^2 + \Delta y^2}$. In such instances, we adopt a straightforward approach: we select the node closest to the desired distance $l$ and establish a connection with it using the chosen link. Importantly, we enforce a one-to-one connection rule, eliminating any instances of multiple links between the same pair of nodes. We repeat this process until we have generated $N\langle k\rangle/2$ links. The resulting link length distribution is practically the same as the target distribution in Eq. (1). Additionally, this method leads to a Poisson degree distribution with mean value $\langle k\rangle$. Since we deal with sparse networks, of densities lower than $10^{-5}$, the value of $\zeta$ does not interfere with the availability of connections. A comparison of empirical distributions with the target distributions is shown in the Supplementary Information, Supplementary Figs. S2 and S3.

## Reporting summary
Further information on research design is available in the Nature Portfolio Reporting Summary linked to this article.

## Data availability
All data generated in this study have been deposited in the Mendeley database and are available on https://data.mendeley.com/datasets/jkvk97nfjc/1[43].

## Code availability
The codes used for numerical simulations and figure plotting in this study are available on GitHub https://github.com/LeyangXue/SpatialKcorePercolation[44].

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

## Acknowledgements
This work was supported by the following funding sources: Israel Science Foundation, Grant No. 189/19 (S.H.); Binational Israel-China Science Foundation, Grant No. 3132/19 (S.H.); NSF-BSF, Grant No. 2019740 (S.H.); EU H2020 Project RISE, Project No. 821115 (S.H.); EU H2020 DIT4TRAM (S.H.); Israel Ministry of Innovation, Science & Technology, Grant No. 01017980 (S.H.); Science Minister-Smart Mobility, Grant No. 1001706769 (S.H.); EU H2020 Project OMINO, Grant No. 101086321 (S.H.); Key Program of the National Natural Science Foundation of China, Grant No. 71731002 (Z.D.); Mordecai and Monique Katz Graduate Fellowship Program (B.G.); China Scholarship Council Program (S.G., L.X.); Israeli Sandwich Scholarship (L.X.); and computational resources supported by the Interdisciplinary Intelligence SuperComputer Center of Beijing Normal University Zhuhai (L.X.).

## Author contributions
L.X. developed the code and performed the numerical simulations. L.X., S.G., L.G., O.L., B.G., Z.D., and S.H. interpreted the results. L.X. drafted the manuscript, L.G. and S.H. revised and improved the manuscript. S.G., O.L., B.G., and Z.D. contributed to the review of the manuscript. S.H. conceived and designed the study.

## Competing interests
The authors declare no competing interests.
