## [Peer Review File · Nature Communications]

REVIEWER COMMENTS

Reviewer #1 (Remarks to the Author):

This manuscript considers a k -core percolation process on a spatial network resembling a lattice with long range links of specified expected length. The key finding is that unlike in other network models, the spatial networks may sustain a cascade of failures initiated by a microscopic event. This process is akin to nucleation but has not been observed in the context of percolation so far and certainly has far reaching implications. In fact, it is quite fascinating that such a simple model exhibits so rich nucleation-like behaviour. I suggest a list of improvements for the manuscript while seeing a clear path to acceptance. My suggestions centre around improving the description of the assumptions that go into the model as well as better crystallising the conclusions.

INTRODUCTION

1) First of all, it would be fair to the reader to mention that in a simple square lattice, when started with a microscopic hole, the k -core percolation can be made to propagate until the whole network is dismantled. (A hole in a square lattice opens a front of vertices with coordination number less than 4, which will eventually expand through the whole system). This is easy to characterise analytically, and can serve as a good toy model to be compared with the findings of this manuscript, such as Supplementary Figure 4.

Of course, the novelty of this manuscript is the existence of a discontinuous jump and the non trivial way in which the characteristic link length affects hole propagation.

MODEL

2) "We fix the value of ζ and generate $N \langle k \rangle / 2$ links with lengths distributed according to $P(l)$ in Eq. 1" — this explanation is slightly misleading, as you first sample link lengths, i.e. real numbers, and then, for each number, you attempt to generate a link starting at a random vertex that approximate this length. Perhaps you can state this more clearly.

Furthermore, this begs a question: how good is this strategy of first selecting a random vertex and then choosing a link as opposed to selecting a link from a list of all potentially available link length? Does it allow you to satisfy empirical $P(l)$ closely to the desired functional shape? Is edge depletion a concern for large $\langle k \rangle$?

3) It is also not clear whether the results converge with N being large. A supplementary figure reporting convergence of $P(l)$, degree distribution, p_c , $P_\infty(p_c)$ would resolve this.

4) The role of the degree distribution is not clear and it is not clear whether the distribution remains Poisson as is claimed in the introduction. Authors manipulate $\langle k \rangle$ as though it is an independent input parameter, but it is not automatically clear whether or not changing ζ also affects the degree distribution. If true, will this change also affect the resilience of the network?

RESULTS

5) Figure 2. The annotation with capital letters A,B and C corresponding to panels, respectively (b),(c),(d) is quite confusing. Also it is not immediately clear that (d) corresponds to two panels while (b) and (c) are single panels. Are the results averaged over many simulations?

6) I feel that this section can be rewritten to be more clear and in the same more concise. For instance, I do not understand what is the mechanism behind the abrupt phase transition. For example, at $\zeta = 10$, Figure 2d shows a hole propagation. How does that lead to a discontinuous jump shown in Figure 1c? Especially given that Figure 3a shows that the giant component is vanishing at the critical point. Furthermore, authors claim that "After the random removal of the initial nodes, a large enough hole emerges spontaneously." — this makes an impression that there can be only one hole. Why?

7) "This means that a microscopic intervention yields a macroscopic phase transition." One has to be careful with this statement, as it can potentially be misleading. Hole's radius propagates linearly in time, t , hence the size of the hole is of the order $O(tL)$, and to dismantle a positive fraction of the system vertices we need t to be of order L . When L diverges to infinity, we would need infinite time to dismantle the system. In this type of cascade we do not have exponential growth and therefore discussing the role of time needs more attention in the manuscript. In other words, one trades a dimension of space for time. I need to stress that in other percolation models, it is customary to assume that a positive fraction of links fails. This is because those fails are independent events. Here, we have a clear temporal dependency and time plays a role of a limiting factor. Another question that pertains to this discussion is the role of dimensionality of the lattice. Will the hole propagate faster in dimension 3?

8) I find Supplementary Figure 3 to be the most interesting in the paper, and I think it deserves to be placed in the main text. Also more investigation on how the link length affect the speed of hole propagation would be insightful.

Reviewer #2 (Remarks to the Author):

The paper concerns the percolation of k -core decomposition in the Euclidean space. This percolation model allows for a long-range connection with a characteristic length scale ζ , the degrees has a Poisson distribution with the mean value $\langle k \rangle$. Each site is occupied with probability p . In the supercritical regime, $p > p_c$, a node is deleted and k -core decomposition is implemented. Depending on the parameters ζ , $\langle k \rangle$, k , the k -core pruning process generates a percolation transition, which can be discontinuous, mixed-order, and continuous. By controlling the parameter ζ , the crossover from random network to Euclidean space is managed. Therefore, the order parameter changes from discontinuous to continuous. The authors claim that a metastable state exists, which induces discontinuity and the system is significantly fragile therein.

I think the results are interesting. However, the authors present only their macroscopic results, but the digging of fundamental cause is missing. For instance, what mechanism leads to the metastable state, why the system is fragile in the metastable state, how a mixed-order is created. The statement “a microscopic intervention yields a macroscopic phase transition” is too exaggerating to mislead the readers. That is just a jump of the order parameter.

Moreover, in the discussion section, the authors mention about the analogy of the current model to the percolation in interdependent networks, and possible existence of a universality class, but unknown yet. However, cascade failures in interdependent network was previously in-depth studied, and its universal behavior has already been uncovered.

Reviewer #1 (Remarks to the Author):

INTRODUCTION

1) First of all, it would be fair to the reader to mention that in a simple square lattice, when started with a microscopic hole, the k-core percolation can be made to propagate until the whole network is dismantled. (A hole in a square lattice opens a front of vertices with coordination number less than 4, which will eventually expand through the whole system). This is easy to characterise analytically, and can serve as a good toy model to be compared with the findings of this manuscript, such as Supplementary Figure 4. Of course, the novelty of this manuscript is the existence of a discontinuous jump and the non trivial way in which the characteristic link length affects hole propagation.

Thank you for suggesting this simple model to better explain the process. This toy model corresponds to the simplified limiting case $\zeta=1$ and $k=4$ (with the absence of randomness in both ζ and k). We have added a short description of this model in the main text, and we have discussed some of the trivial analytical results in the Supplementary Information for better understanding the model we present in this paper.

MODEL

2) "We fix the value of ζ and generate $N\langle k \rangle/2$ links with lengths distributed according to $P(l)$ in Eq. 1" — this explanation is slightly misleading, as you first sample link lengths, i.e. real numbers, and then, for each number, you attempt to generate a link starting at a random vertex that approximate this length. Perhaps you can state this more clearly.

Furthermore, this begs a question: how good is this strategy of first selecting a random vertex and then choosing a link as opposed to selecting a link from a list of all potentially available link length? Does it allow you to satisfy empirical $P(l)$ closely to the desired functional shape? Is edge depletion a concern for large $\langle k \rangle$?

Thank you for pointing this distinction. In the revised manuscript, we clarify the method used to distribute the links in the system and we believe that the explanation is now more accurate. We had also verified in the revised manuscript that the empirical distribution is practically the same as the desired distribution. We have now added a plot in the Supplementary Information of the revised manuscript (Fig. S2, also reproduced below, as Fig. RFig1) which clearly shows that the two distributions are almost identical.

RFig1. The probability densities of the link length, $p(l)$, and the degree, $p(k)$, are not influenced by our network construction method. The link length distribution, $p(l)$, is shown for different values of ζ , for (a) $\langle k \rangle = 5$ and (c) $\langle k \rangle = 10$. Similarly, the resulting degree distribution, $p(k)$, remains close to the Poisson distribution with parameter λ , represented with a solid line, for (b) $\langle k \rangle = 5$ and (d) $\langle k \rangle = 10$.

3) It is also not clear whether the results converge with N being large. A supplementary figure reporting convergence of $P(l)$, degree distribution, p_c , $P_{\infty}(p_c)$ would resolve this.

We are grateful to the reviewer for this comment. While we had tested for convergence and $L=500$ ($N=250000$) seemed to work well in the cases that we had studied, following the referee's remark we extended the range of model parameters and realized that to cover all cases we need to increase the size to $L=1000$ ($N=10^6$). This is shown in the new Fig. S3, added to the revised supplementary information (figure reproduced below, as Fig. RFig2), which is one of the cases where convergence occurs at larger sizes. For consistency, this observation led us to modify Figures 1 and 2 in the main text of the revised manuscript and use the results of system sizes of $L=1000$. The numerical values of the results only changed very slightly without influencing the behavior, so that all previous results and conclusions are still valid.

RFig2. The effect of system size on the model and on percolation properties. (a) The exponential distribution of link lengths is not influenced by the system size. **(b)** The degree distribution remains invariant on system size changes and λ remains the same. **(c)** The value of the critical point, p_c , converges at $L=1000$. **(d)** The fraction of nodes in the giant connected component at the critical point, $P_\infty(p_c)$, also converges at $L=1000$. All results are obtained for a network with $\langle k \rangle = 7.5$ and $\zeta = 10$. Each point in (c) and (d) represents the average over 300 realizations for $L < 1000$, 100 realizations for $L = 1000$, and 50 realizations for $L > 1000$, with the error bars indicating the standard deviation.

4) The role of the degree distribution is not clear and it is not clear whether the distribution remains Poisson as is claimed in the introduction. Authors manipulate $\langle k \rangle$ as though it is an independent input parameter, but it is not automatically clear whether or not changing ζ also affects the degree distribution. If true, will this change also affect the resilience of the network?

Thank you very much for this comment. Following this comment, we tested this issue in the revised manuscript and found indeed, that the value of ζ does not modify the Poisson degree distribution. This can be seen in Figs. S2 and S3 added to the revised manuscript and shown above, where $P(k)$ remains invariant for values of ζ in the range from $\zeta=4$ to $\zeta=100$.

In general, our results refer to sparse networks, since the maximum value we use for average degree is $\langle k \rangle = 8$ in a system of $N = 10^6$, so that the network density is $\langle k \rangle / (N-1) \sim 10^{-5}$. This low density leads to a statistically high probability of locating available connections without rejecting any significant number of links. Except for certain combinations of extreme values which restrict the links within a narrow neighborhood, e.g. very small ζ and very large $\langle k \rangle$, the value of ζ does not interfere with the availability of connections. We discuss this issue in the revised manuscript.

RESULTS

5) Figure 2. The annotation with capital letters A,B and C corresponding to panels, respectively (b),(c),(d) is quite confusing. Also it is not immediately clear that (d) corresponds to two panels while (b) and (c) are single panels. Are the results averaged over many simulations?

We thank the reviewer for this comment, which helped us to significantly improve Figure 2 and its readability in the revised manuscript. The reviewer is correct that the figure could be confusing. We have now rearranged the panels in the figure by clearly pointing out the values which correspond to each panel and added arrows to visually guide the reader. We believe that it is now much easier to follow the figure.

Since the purpose of the panels is to visually illustrate the network structure, they necessarily depict one specific realization, which is indicative of the structure for the given parameters. The results in this plot are averaged over 300 realizations. All the plots in the paper are averaged over 50 to 1000 realizations, depending on the model parameters. We now explicitly mention the number of realizations in each figure caption of the revised manuscript.

6) I feel that this section can be rewritten to be more clear and in the same more concise. For instance, I do not understand what is the mechanism behind the abrupt phase transition. For example, at $\zeta = 10$, Figure 2d shows a hole propagation. How does that lead to a discontinuous jump shown in Figure 1c? Especially given that Figure 3a shows that the giant component is vanishing at the critical point. Furthermore, authors claim that "After the random removal of the initial nodes, a large enough hole emerges spontaneously." — this makes an impression that there can be only one hole. Why?

We thank the reviewer for identifying unclear statements regarding the mechanism in the explanation of the results in the original manuscript. We have now revised the discussion in the manuscript to better clarify the mechanism behind the transition and the existence of just one large-scale hole under certain conditions.

The main questions, as stated by the referee, are (i) how a hole propagation mechanism similar to former Fig.2d can lead to the abrupt transition shown in Fig. 1c, and (ii) why there is only a single propagating hole when the breakdown occurs. To answer these questions, we need to quickly summarize our model. After preparing the network with parameters ζ and $\langle k \rangle$, we remove a fraction of nodes, $1-p$, chosen randomly. We then allow the system to evolve spontaneously by iteratively removing nodes whose k -shell value is less than the defined threshold. When the process ends and no more nodes can be disconnected due to the k -core process, we record the value of the giant component size $P_{\infty}(p)$. Notice that this is an *equilibrium measurement*, and its value depends only on the initial fraction of removed nodes, $1-p$, without any direct dependence on the iterations that occur spontaneously to reach this value. The abrupt transition in Fig. 1c shows that the giant component size, $P_{\infty}(p)$, still exists above

p_c , but just below p_c , the cascading is catastrophic and yields the abrupt collapse. Thus, the steady state solutions change discontinuously when we cross p_c , representing the critical threshold for the fraction of *initially removed nodes*, $1-p$. Notice that in the area above the curve in Fig. 2a, the system is stable, and the giant component has a finite value because the fraction of initially removed nodes is small and cannot trigger the network collapse. The curve represents the critical points, p_c , where k-core percolation becomes strong enough to spontaneously destroy the entire system and this is also the case for all points below the curve.

Having established the existence of the abrupt transition in the intermediate values of ζ , we seek to understand the mechanisms which lead to this catastrophic collapse of $P_{\infty}(p)$ as we vary p . To do that, we need to follow the 'time' evolution of the k-core percolation cascading process so that we can observe how the system disintegrates. Notice that 'time' is only used here as a convenient measure to follow the system evolution, and in principle we could assume that nodes fail immediately when their current degree falls below the threshold. To follow the cascading mechanisms, here we define that each iteration of removing nodes corresponds to one time-step. In the bottom row of Fig. 2a (visually) and Fig.3 (quantitatively) we track the *microscopic* changes of the system as a function of these 'time steps', where each time step represents one k-core pruning iteration, i.e. a cascading step, as shown in Fig. 1a. The panels in Fig.2a show that in the intermediate ζ range a nucleation process emerges, where a hole is formed spontaneously somewhere in the system (which is a spatial regime of very low density region - due to the random removal) and then radially increases with time, eventually covering the entire system. The radial propagation of the initial damage is caused by the typical length of links, ζ , which represents the maximal distance of the next step failures from the surface of the earlier step hole. This means that nodes at distance ζ will lose links and their degree could go below k and therefore removed. This spontaneous propagation ends when the hole reaches the boundary, thus answering question (i). We now included the above explanation in the revised manuscript.

For answering question (ii) of why there is only a single hole that propagates, we added in the revised version the following argument. The single hole appears due to the random spatial disorder in the system just before the catastrophic cascade occurs. Once a small hole emerges anywhere in the system the ζ links enable its propagation, and this expanding hole causes the abrupt nucleation transition. Indeed, our simulations show (see a typical example at the bottom row of new Fig. 2a) a dominant hole that takes over the system and propagates yielding a system collapse.

As a demonstration of this abrupt transition, a similar cascading behavior can be seen in the toy model of a full square lattice as suggested by the reviewer. In this case the initial fraction of nodes in the giant component is 1. If we remove a single node anywhere in the lattice the fraction of nodes remains 1 (in an infinite system) but then after each $k=4$ -core pruning step the cascading process removes another internal layer leading to a growing hole which at the equilibrium will lead to the absence of a giant component. In other words, the removal of just one node leads to a discontinuous transition via the emergence of one giant hole.

In short, the abrupt transition of the giant component size as we lower p is because at large p values very few nodes are removed by the k-core percolation process, but when we reach a critical value, p_c , there are enough nodes removed to lead to the emergence of a hole somewhere in the lattice which is driven to propagate by the ζ links and keeps increasing, so

that it gradually consumes the entire system, destroying any large-scale connectivity and causing the abrupt collapse.

In the revised manuscript we discuss the above insights that were not clearly stated in the earlier version.

7) “This means that a microscopic intervention yields a macroscopic phase transition.” One has to be careful with this statement, as it can potentially be misleading. Hole's radius propagates linearly in time, t , hence the size of the hole is of the order $O(tL)$, and to dismantle a positive fraction of the system vertices we need t to be of order L . When L diverges to infinity, we would need infinite time to dismantle the system. In this type of cascade we do not have exponential growth and therefore discussing the role of time needs more attention in the manuscript. In other words, one trades a dimension of space for time. I need to stress that in other percolation models, it is customary to assume that a positive fraction of links fails. This is because those fails are independent events. Here, we have a clear temporal dependency and time plays a role of a limiting factor. Another question that pertains to this discussion is the role of dimensionality of the lattice. Will the hole propagate faster in dimension 3?

We thank the reviewer for pointing these questions since it helped us to clarify better the mechanisms in the revised manuscript. The reviewer is correct that we should have discussed time more carefully in our manuscript. We have now modified the text, based on the following considerations.

As discussed in answering comment 6, in our work we consider time as an *artificial* way of keeping track of removed nodes and of the cascading process. In practice, we can consider that we perturb the system by initially removing a set of nodes and observe the system evolving through the k -core percolation process until it reaches the steady state. The use of time steps is a convenient way to monitor the mechanism which drives the system to this final steady state. This is similar to the toy model of a simple square lattice, suggested by the reviewer, and using the same definition of time, it would take an infinite time to destroy the network, even though we know that there will be only one hole which will keep expanding. Thanks to the reviewer we now better explain in the revised manuscript the concept of “time”.

According to the above discussion, our statement “a microscopic intervention yields a macroscopic phase transition” could be understood, even though it could be confusing. We have now clarified this statement in the text in the following way. By “microscopic intervention” we mean that we remove a fraction of nodes whose measure is 0. This is justified by e.g. Fig. 4c and 4d, where the critical hole size remains constant independently of the system size and as the system size grows to infinity, this critical hole has a fraction equal to 0. The toy square lattice model (suggested by the referee) could explain this well, since the removal of one node represents the minimum intervention possible. This microscopic removal leads a macroscopic system to collapse as completely disjointed, an effect which is observed macroscopically, i.e. at the level of the global system. We also explain accordingly the statement “a microscopic intervention yields a macroscopic phase transition” in the revised manuscript.

We currently do not have any arguments on whether the nucleation process in dimension=3 would be faster, but we believe that this would be a very interesting topic for future studies.

8) I find Supplementary Figure 3 to be the most interesting in the paper, and I think it deserves to be placed in the main text. Also more investigation on how the link length affect the speed of hole propagation would be insightful.

Thank you for this comment. Accordingly, we have rearranged the panels in the figures, by incorporating parts of former Fig. S3 in Fig. 2. As mentioned above, the radial propagation of the initial damage is caused by the typical length of links, ζ . This ζ value represents the maximal distance of the failures which occur at the next step from the surface of the current hole, so that longer link lengths lead to higher speeds, as shown in Fig. 3 and Fig. 2b (which was part of Fig. S3 in the previous version).

Reviewer #2 (Remarks to the Author):

I think the results are interesting. However, the authors present only their macroscopic results, but the digging of fundamental cause is missing. For instance, what mechanism leads to the metastable state, why the system is fragile in the metastable state, how a mixed-order is created. The statement “a microscopic intervention yields a macroscopic phase transition” is too exaggerating to mislead the readers. That is just a jump of the order parameter.

We thank the reviewer for pointing out the confusion caused by our description. There are two main questions in this remark: (i) What is the fundamental cause of the macroscopic results, i.e. what are the microscopic mechanisms which lead to the different results, such as the metastable state and the mixed-order transition, and (ii) is the statement “a microscopic intervention yields a macroscopic phase transition” valid?

For the first question, we have devoted in the revised version an extensive part of the Results and Discussion sections to describe how different values of ζ give rise to different microscopic mechanisms and we discuss how these mechanisms eventually lead to different types of steady states and types of transition. The main points are as follows:

(a) The metastable state is a result of the system structure, as generated by the values of $\langle k \rangle$ and ζ . The stability of the system is determined by the effect of a localized attack. If the system collapses by a localized attack of any size, then the system is unstable. If the system remains connected after any localized attack, then the system is stable. In this study, we find that certain combinations of $\langle k \rangle$ and ζ , however, render the system metastable, so that the size of the initial hole determines whether the system will remain connected or if it will collapse, as shown in Fig. 4b. The critical size of the hole, r_h^c , above which the system collapses, is independent of the system size and therefore can be regarded as microscopic. It only depends on $\langle k \rangle$ and ζ , which shows that the underlying mechanism is determined by the system topology only. After generating the initial hole of size above r_h^c , the links of size ζ enable the propagation of the hole radially via cascading failures causing the system collapse. The behavior of the different phases is shown in the phase diagram of Fig. 4.

(b) The mixed-order transition emerges at large ζ values, of the order of the system size, L . In this range, the system becomes more homogeneous because the majority of the links are long range and there is no nucleation, yielding mixed-order transitions. In this case a critical plateau occurs near p_c , as seen in Fig. 3b in contrast to the nucleation seen in Fig. 3a.

(c) As for the answer to question (ii), in the metastable state a small localized attack below r_h^c remains localized and does not destroy the system, while a hole above a threshold size will lead to a complete system collapse. This threshold radius is independent of the system size, so that an attack of zero measure, that is of microscopic size, can destroy a macroscopic system. This consideration justifies the statement " a microscopic intervention yields a macroscopic phase transition".

(d) The order parameter describes a macroscopic property of the system. The jump of the order parameter demonstrates a change in the state of the system which can be observed from outside, i.e. it is a macroscopic change in the global state of the system. The cause of this change, though, is the removal of a single node or a very small number of nodes, i.e. a small perturbation which can only be observed at the microscopic level. So, we believe that our statement is accurate and the reason behind this jump in the order parameter is what makes this transition interesting.

We have modified the manuscript to clarify the above comments.

Moreover, in the discussion section, the authors mention about the analogy of the current model to the percolation in interdependent networks, and possible existence of a universality class, but unknown yet. However, cascade failures in interdependent network was previously in-depth studied, and its universal behavior has already been uncovered.

We thank the reviewer for raising this issue that helped us to clarify it better in the revised manuscript. We agree that a different problem, but with analogous behavior, has been studied in the literature on cascading failures in interdependent networks. We are sorry if it sounds, in the original manuscript, like we claim that the behavior of percolation is unknown in interdependent networks. We clarified in the revised manuscript, that our main hypothesis is the possible existence of a *common universality class across different systems*, which can describe percolation a) in interdependent networks, and b) in spatial k-core systems, and c) possibly in other - currently unknown - systems. Similar to classical percolation where a universality class depends only on the system symmetries and dimensionality, we pose the question of whether there is a class of network systems whose percolation properties and critical exponents may depend on features, such as two types of interactions, which may be universal but not yet determined.

REVIEWERS' COMMENTS

Reviewer #1 (Remarks to the Author):

After carefully reading the manuscript and the authors responses, my opinion is that the manuscript has been much improved. The main claims have been crystallised and better supported. The main findings are rather intriguing observations that open grounds for followup research. I agree with Reviewer 2 that it would have been very insightful to have a mechanistic explanation behind the observed phenomena, however it is probably too much to expect this in one paper.

This is not a very critical remark, but my impression is that the text does not read very smoothly. This is probably the result of many edits and having multiple coauthors working on the text. My impression is that authors could have made the text more concise and focus on the main claims. My suggestion would be that one of the coauthors reads through the text again just focusing on the structure, optimising the flow of the story.

Please also check: there is something wrong with the sentence in the Discussion section: "where microscopic intervention results in a —it macroscopic phase transition. "

Please check the link to the dataset from GitHub readme

Reviewer #2 (Remarks to the Author):

The manuscript has been well revised according to my comment. Thus, I have no further comment on it.

Reviewer #2 (Remarks on code availability):

no comment

Reviewer #1 (Remarks to the Author):

This is not a very critical remark, but my impression is that the text does not read very smoothly. This is probably the result of many edits and having multiple coauthors working on the text. My impression is that authors could have made the text more concise and focus on the main claims. My suggestion would be that one of the coauthors reads through the text again just focusing on the structure, optimising the flow of the story.

We thank the reviewer for this constructive comment. We carefully went through the text and made many small improvements, mainly related to the clarity of the text. We changed the structure of the paper by removing parts of the results into the Method section. This hopefully makes smoother the flow of the paper. At the end of each section, we now summarize the main results of that section, when such a summary was missing, thus improving the readability and understanding of the manuscript.

Please also check: there is something wrong with the sentence in the Discussion section: "where microscopic intervention results in a —it macroscopic phase transition."

Thank for pointing out this typo. We have now fixed it by removing the "it" directive which was a mistyped latex command.

Please check the link to the dataset from GitHub readme.

We checked the link and verified that it is now correctly pointing to the dataset.

Reviewer #2:

The manuscript has been well revised according to my comment. Thus, I have no further comment on it.

Thank you for your help with improving our manuscript and for your positive evaluation of the resubmitted manuscript.